# Concentric Spherical GNN for 3D Representation Learning

## Abstract

Learning 3D representations of point clouds that generalize well to arbitrary orientations is a challenge of practical importance in problems ranging from computer vision to molecular modeling. The proposed approach is based on a concentric spherical representation of 3D space, formed by nesting spatially-sampled spheres resulting from the highly regular icosahedral discretization. We propose separate intra-sphere and inter-sphere convolutions over the resulting concentric spherical grid, which are combined into a convolutional framework for learning volumetric and rotationally equivariant representations over point clouds. We demonstrate the effectiveness of our approach for 3D object classification, and towards resolving the electronic structure of atomistic systems.

## 1 Introduction

3D point cloud data appear in domains ranging from computer vision, geographic information systems, and molecular modeling. Learning suitable representations of point clouds for data-driven modeling is well-motivated by applications like automonous vehicles and molecular simulation. It is important and yet challenging to learn representations that generalize well to arbitrary orientations of point clouds, in an efficient and accurate manner. Spherical Convolutional Neural Networks (CNNs) were introduced in Cohen et al. (2018) and Esteves et al. (2018) to address the challenge of rotations for spherical images, by defining rotationally equivariant convolutions in harmonic-space. Convolutions equivariant to transformations (such as translation) underpins recognizable aspects of 2D CNNs such as effective filter sharing and feature localization, a key reason for their success Cohen & Welling (2016). However, Spherical CNNs operate over data projected onto the spherical manifold, which is lossy for general 3D point cloud data. It is desirable to learn features volumetrically, in order to accurately detect patterns in complex 3D shapes, or incorporate spatial relationships between atoms in describing atomic environments. We show that it is more expressive and general to operate over an underlying spatial representation of concentric spheres, demonstrated experimentally.

We propose a new spatial structure consisting of multiple nested spheres, each discretized by the icosahedral grid. This structure is used to record both the angular and radial distribution of input points, as volumetric information. The icosahedral grid produces a highly regular sampling of the sphere, which permits efficient convolutions scaling linearly with spherical resolution, explored in Jiang et al. (2019), Cohen et al. (2019), and Defferrard et al. (2020). We then propose two types of convolution for learning over the concentric spherical structure, by separately learning intra-sphere and inter-sphere features. We formulate intra-sphere convolution in terms of graph-based convolution over localized intra-sphere neighborhoods, and inter-sphere convolution as 1D convolution over co-radial grid points. The resulting convolutions are both rotationally equivariant and scalable, each scaling (near) linearly with respect to the size of the grid. Finally, we incorporate the proposed convolutions into a hierarchical and multi-resolution architecture, CSGNN, for learning over concentric spherical feature maps, and demonstrate its applicability to point cloud data.

We first apply our approach to the problem of classifying rotated 3D objects sampled by point clouds, achieving state-of-the-art performance on the task. We further apply our approach to a problem of molecular modeling, namely predicting the electronic density of states of materials. The density of states is a fundamental property of electronic structure, used in determining total energy contributions. Our approach is applied to learn localized descriptions of atomic environments, enabling more

accurate resolution of the band energy of carbon-based materials compared to previous approaches. The implementation of our methods and experiments are publicly available [1].

In summary, our primary contributions are as follows:

1. We propose a new volumetric representation as the basis for convolutional learning. This representation consists of multiple nested spheres, each discretized by the icosahedral grid.
2. We introduce a novel architecture for learning volumetric representation over concentric spheres by combining intra-sphere and inter-sphere convolutions. The proposed convolutions are rotationally equivariant, and also scale (near) linearly with grid resolution.
3. We demonstrate the applicability of our approach through experiments in 3D object classification, and resolving electronic structure of atomistic systems.

## 2 BACKGROUND AND RELATED WORK

The goal of learning representations of general 3D point cloud data has led to a diverse body of work for structured learning (Maturana & Scherer, 2015; Qi et al., 2017a;b; Wang et al., 2019; Thomas et al., 2019; Zhang et al., 2019b). However, the main problem shared by these methods is that they do not generalize well to general rotations of the data, which can lead to catastrophic loss of performance when they are encountered. Augmenting training with rotated data helps bridge the gap somewhat, but a significant performance gap remains. The key missing piece in many earlier work is their lack of rotationally equivariant model design. A model layer is *equivariant* to rotation if it commutes with rotation. In other words, feeding a rotated input to the model layer is same as feeding the original input to the layer and rotating its output. A rotationally *invariant* layer is a special case of an equivariant layer where rotation does not affect the output space of the layer. Several models have been proposed which exclusively use rotationally invariant layers (Schütt et al., 2017; Gilmer et al., 2017; Schütt et al., 2017; Chen et al., 2019; Zhang et al., 2019a; Poulenard et al., 2019; Kim et al., 2020). However, using invariant layers through the entirety of model is unnecessarily restrictive, as important information about the underlying spatial structure of the data may be lost. Our work focuses on designing rotationally equivariant layers as primary building blocks, while invariant layer(s) can be used before final output to achieve overall invariance of the model. This design has already seen extension to other structures and learning strategies relevant to point cloud representation. Thomas et al. (2018) proposed rotation equivariant point-wise convolutions for 3D graphs, but their approach has difficulty scaling to point clouds beyond the scale of small molecules. Equivariant design has also seen extension to spherical images through Spherical CNNs (Cohen et al., 2018; Esteves et al., 2018; Jiang et al., 2019; Cohen et al., 2019; Rao et al., 2019; Defferrard et al., 2020; Yang et al., 2020). However, they are not well-suited for direct application to point clouds. Their key limitation is loss of information in constraining spatial representation from 3D domain to a 2D (spherical) manifold. Our work overcomes this limitation by proposing convolutional learning over concentric spheres, achieving rotationally equivariant 3D feature learning with scalable convolutions.

## 3 ARCHITECTURE DESIGN

The primary goal of our proposed approach is to learn volumetric representations of 3D point clouds in a rotationally equivariant and also scalable manner. To achieve this goal, we propose concentric spheres at different radii, each discretized by the icosahedral grid. The proposed construction naturally organizes 3D data by angular and radial distribution, where the resolution of each component can be controlled independently. We use the icosahedral grid as it results in a highly regular sampling of the sphere. The former permits efficient use of spatial resolution, and the later results in design of efficient and rotationally-equivariant convolutions. We propose using two separate convolutions together to learn volumetric features over concentric spheres: (1) graph-based convolution to incorporate information within spheres, and (2) radial convolutions to separately incorporate information between spheres. The proposed convolutions are extended to different spatial scales via pooling based on regular properties of the icosahedral grid, resulting in the proposed hierarchical convolutional architecture of Fig. 1. We explain each component of our model in more detail in subsequent sections, and pooling in A.5 of Appendix.

---

[1] https://github.com/anonymous10521/CSGNN

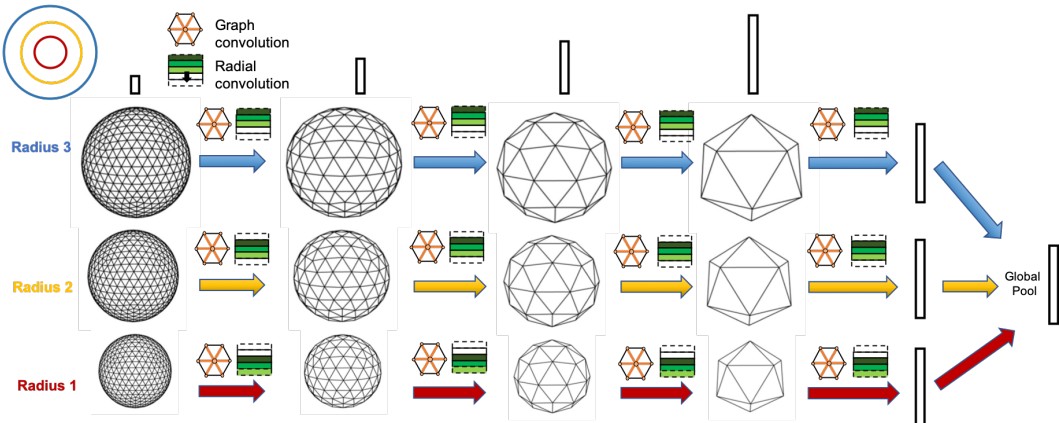

Figure 1: Example architecture with three concentric spheres. Graph convolutions are followed by radial convolutions, at each level of spherical resolution. Radial convolution (in this example) has spatial window of three co-radial vertices, with padding applied to maintain radial dimensions across convolutions. Each arrow indicates vertex neighborhood pooling and downsampling, after which convolutions proceed with new filters at coarser spatial resolution. Global pooling is applied to obtain final feature representation.

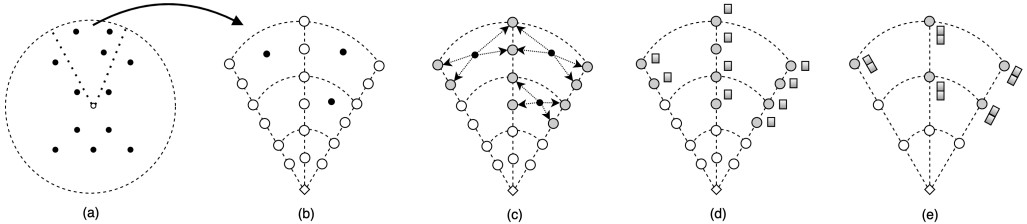

Figure 2: Illustrative example of converting point cloud to input features. (a) shows an example point cloud (black points) contained within a bounding sphere. (b) The spherical volume is partitioned into 6 concentric spheres, co-radially. (c) Each point has a contribution to vertices in a local neighborhood (gray circles), resulting in (d) single channel feature per vertex (gray square). (e) Co-radial vertices are further grouped, resulting in smaller subset of concentric spheres with multi-channel inputs. In this example, grouping results in 3 concentric spheres where vertices have 2 input channels each.

## 3.1 CONCENTRIC SPHERICAL DISCRETIZATION AND POINT CLOUD MAPPING

In this section we explain in detail our method of volumetric discretization by concentric spheres. We further present our approach to converting arbitrary point cloud data to initial feature channels over this spatial structure.

**Concentric Icosahedral Spheres.** The initial icosahedron has 12 vertices forming 20 equilateral triangular faces. To increase grid resolution, each face can be sub-divided, with resolution scaling as $|V| = 10 * 4^l + 2$ ($l$ is target discretization level). See Fig. 3a for an illustration of this process. We implement concentric spheres by stacking $R$ identical icosahedral grids to form the radial dimension, shown in Fig. 3b. Assuming normalization to unit radius, we uniformly assign concentric spheres at radii $[\frac{1}{R}, \frac{2}{R}, ..., 1]$. Assuming single-channel feature map, the resulting grid is the matrix $\boldsymbol{H} \in \mathbb{R}^{R \times |V|}$, where each vertex is indexed by the sphere it belongs to, and its position on the sphere. The resulting volumetric representation has several noteworthy properties. First, the icosahedral discretization results in a highly regular spatial sampling within each sphere. This allows more efficient use of spatial resolution compared to polar grids used in earlier works (Cohen et al., 2018; Esteves et al., 2018; You et al., 2020), which have have resolution bias towards the polar regions. Second, spatial resolution is not uniform across different spheres, as the proposed construction results in higher sampling density closer to the center. However, this non-uniformity difference is largely accounted for as a function of the radius of the sphere, and does not inhibit the design of efficient and rotationally equivariant convolutions.

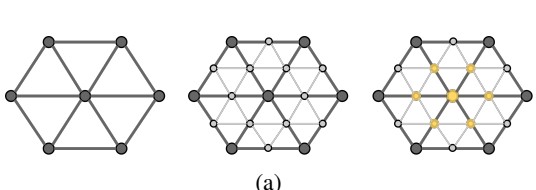 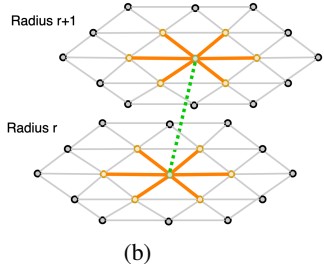

Figure 3: (a) The icosahedral grid is formed by vertices of equilateral triangles (left), which can be recursively sub-divided to form a higher resolution grid (middle). This also defines a natural vertex neighborhood and hierarchy for pooling and downsampling, where yellow highlighted vertices (right) are involved in pooling. (b) Two spherical grids are stacked, corresponding to consecutive concentric spheres. Graph convolution involves vertices within the same sphere (convolution neighborhood highlighted orange). Conversely, radial convolution involves co-radial vertices between the two spheres (green dotted line).

**Point Cloud to Concentric Spheres.** We now consider the problem of describing a point cloud $\boldsymbol{P} \in \mathbb{R}^{N \times 3}$ in terms of concentric spherical feature map $\mathsf{H} \in \mathbb{R}^{R \times |V| \times C}$, where $C$ is number of channels. While the concentric grid representation is defined discretely, the space point positions are continuous, which we would like to capture. To do so, we summarize the contribution of points using the Gaussian radial basis function (RBF):

$$f(\boldsymbol{x}) = \sum_{j=1}^{N} \phi(||\boldsymbol{x} - \boldsymbol{P}_j||_2^2) \tag{1}$$

$N$ is the number of data points, and the function $\phi = \exp(-\gamma r^2)$ is parameterized by the bandwidth $\gamma$. In practice we limit computation to a local neighborhood (instead of considering all points) and choose $\gamma$ accordingly. We refer to Fig. 2 for illustrative example of the conversion, and Sec. A.6 for implementation details as applied in experiments.

We determine the input channels $C$ by grouping concentric spheres and concatenating features of co-radial vertices, also shown in Fig. 2. Suppose converting the point cloud results in a single-channel feature tensor $\mathsf{H}' \in \mathbb{R}^{R' \times |V|}$, where $R'$ is the initial number of spheres. Grouping the features from co-radial vertices across $R$ groups results in feature tensor $\mathsf{H} \in \mathbb{R}^{R \times |V| \times \frac{R'}{R}}$, where $R$ is the number of spheres represented spatially, and $\frac{R'}{R}$ is the number of spheres represented via input channels. The proposed grouping mechanism gives flexibility to significantly increase radial resolution without correspondingly increasing the size of the spatial grid, which persists across convolutions. This enables balancing between computational efficiency and the representational boost of having additional spheres.

## 3.2 CONCENTRIC SPHERICAL CONVOLUTIONS

In this section we present our implementation of rotationally equivariant intra-sphere and inter-sphere convolutions for learning volumetric features. Proof of equivariance is provided in Sec. A.1.

**Intra-sphere convolutions.** The objective for intra-sphere convolutions is to learn localized features within each sphere, in a rotationally equivariant fashion. We use graph convolutional filters for this objective, as localized graph convolutions on icosahedral grids are rotationally equivariant with respect to the icosahedral rotation group. This motivates our construction of the undirected graph $G^{(l)} = (V^{(l)}, E^{(l)})$ from level $l$ icosahedron $I^{(l)}$. Vertices of the vertex set $V^{(l)}$ correspond one-to-one with vertices of $I^{(l)}$, projected to unit sphere, and $E^{(l)}$ is the set of edges corresponding to face edges of the icosahedron. We refer to Fig. 3b for an example of graph connectivity over each sphere. The resulting graph is highly regular, as all vertices are degree six beyond $I^{(0)}$. All edges within each sphere are also approximately equidistant (Wang & Lee, 2011). We use the graph convolutional operator from Kipf & Welling (2017), but omit the degree-based normalization due to the regularity of our graph construction. We introduce additional notation to define graph convolution in the context of the proposed discretization. Let $\mathsf{H} \in \mathbb{R}^{R \times |V| \times C}$ denote a $C$ channel tensor of features, and $\boldsymbol{Z} \in \mathbb{R}^{C \times F}$ be learnable weights, where $C$ and $F$ are input and output channel sizes.

We use to $N(u)$ denote neighbors of vertex $u$ in graph $G$. We also introduce subscript $t$ to indicate convolutional layer number, $i \in [0, R-1]$ to index the radial dimension, and $u \in [0, |V|-1]$ to index the vertices. The layer $t+1$ intra-sphere convolution output for vertex $u$ of sphere $i$ is then given by Eq. 2, where $\sigma$ is a nonlinear activation function:

$$\mathbf{H}_{i,u}^{(t+1)} = \sigma(\sum_{v \in N(u)} \mathbf{H}_{i,v}^{(t)} \boldsymbol{Z}^{(t)}) \tag{2}$$

**Inter-sphere convolutions.** We introduce *radial convolutions* to learn features between spheres. To do so, we propose applying 1D convolution over co-radial vertices. In treating co-radial vertices as a sequence, 1D convolution is intended to learn localized features across spheres while distinguishing their relative positions. Importantly, the proposed convolution is rotationally equivariant, as co-radial vertices transform identically under rotation. We refer to Fig. 3b for illustration. We introduce some additional notation to describe radial convolutions: let $K$ be the size of the 1D convolution kernel window. We pad inputs in the radial dimension such that the number of spheres $R$ is maintained spatially across convolutions. Let $\boldsymbol{W} \in \mathbb{R}^{K \times C \times F}$ be a tensor of shared parameters, where $C$ and $F$ are input and output channel sizes. The layer $t+1$ radial convolution output for vertex $u$ of sphere $i$ is:

$$\mathbf{H}_{i,u}^{(t+1)} = \sigma(\sum_{k=-\lfloor \frac{K}{2} \rfloor}^{\lfloor \frac{K}{2} \rfloor} \mathbf{H}_{i+k,u}^{(t)} \mathbf{W}_{k+\lfloor \frac{K}{2} \rfloor}^{(t)}) \tag{3}$$

### 3.3 COMPLEXITY ANALYSIS

The neighborhood size of both graph and radial convolutions are constant, and so are their filter parameters due to weight sharing. Therefore the overall complexity of both the graph convolution and radial convolution is $O(R \times |V|)$, or linear with respect to the total discretization size. This introduces an additional factor $R$ of computational and memory cost compared to the $O(|V|)$ complexity of some spherical CNNs, corresponding to stacking multiple spherical discretizations. However, as we show in later experiments, the number of spheres can be small in practice while providing considerable performance benefit. We further provide empirical runtime analysis with respect to $R$ in Sec. A.4 of Appendix.

## 4 EXPERIMENTS

We demonstrate the effectiveness of our approach for 3D object classification in Sec. 4.1, and applied to resolving fundamental properties of electronic structure in materials in Sec. 4.2. We further study the main components to the performance of our proposed approach in Sec. 4.3.

### 4.1 POINT CLOUD CLASSIFICATION

We consider the ModelNet40 3D shape classification task, with 12308 shapes and 40 classes. We use the pre-processed point clouds from Qi et al. (2017a), and 1024 points per point cloud. In total, 9840 shapes are used for training and 2468 for testing.

**Architecture and Hyperparameters.** See Fig. 4 for overview of model layers and components. Batch normalization and ReLU activation is applied after each convolution and hidden layer. A residual connection is also added between each graph convolution layer when the number of input and output channels match. The point cloud is converted to initial concentric spherical features following procedure detailed in A.6. We follow prior work in augmenting training with random translation, re-scaling, and positional jitter of input point clouds. We trained separate models when training with $z$-aligned vs. $SO3$ rotations, and refer to Sec. A.2 for hyperparameter and additional experiment details.

**Results.** For experimental evaluation, we consider two types of rotations in training and/or testing, following convention from earlier work: $z$-axis aligned rotations and arbitrary rotations ($SO3$). The $SO3$ case is most challenging, as it indicates objects can be in any rotational orientation. We compare with related work organized into three categories, based on their strategy for handling

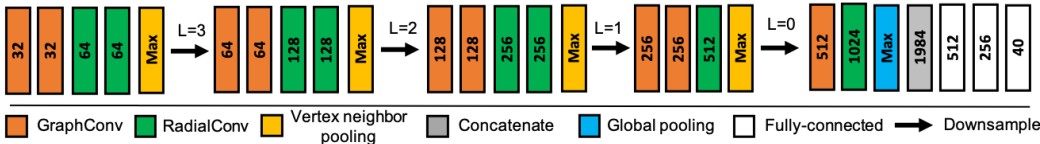

Figure 4: Architecture for ModelNet40 classification. Number of output channels are shown for each layer, where applicable. $L = 4$ is initial level of discretization of icosahedral spheres. Radial convolution uses kernel size of 3 for spatial size (co-radial vertices).

| Method | Strategy | Params | z/z | z/SO3 | SO3/SO3 |
|---|---|---|---|---|---|
| PointNet (Qi et al., 2017a) | Augmentation | 3.5M | 87.5 | 22.9 | 84.9 |
| DGCNN (Wang et al., 2019) | Augmentation | 1.8M | 90.7 | 35.5 | 89.0 |
| ShellNet (Zhang et al., 2019b) | Augmentation | 470K | 89.2 | 22.9 | 84.8 |
| KPConv (Thomas et al., 2019) | Augmentation | 6.1M | 90.0 | 27.5 | 85.0 |
| SPHNet (Poulenard et al., 2019) | Invariance | 2.9M | 86.5 | 85.6 | 87.0 |
| RIConv (Zhang et al., 2019a) | Invariance | 0.7M | 87.0 | 87.0 | 87.2 |
| RI-GCN (Kim et al., 2020) | Invariance | 4.4M | 89.2 | **89.3** | 89.1 |
| SFCNN (Rao et al., 2019) | Equivariance | 9.2M | 90.8 | 84.2 | 89.6 |
| PRIN (You et al., 2020) | Equivariance | 1.7M | 76.5 | 81.9 | 81.0 |
| Ours (CSGNN) | Equivariance | 4.0M | **91.0** | 88.3 | **90.1** |

Table 1: ModelNet40 object classification overall accuracy, considering two types of rotations: $z$-axis aligned, and more general $SO3$ rotations. For example, $SO3/SO3$ indicates training and testing with arbitrary rotations of input data. Strategy refers to how rotations are handled. Number of parameters is in millions.

rotations. Resulted are presented in Table 1, where we ran each baseline. Our method achieves state of the art performance in z/z and SO3/SO3 settings when training and testing from same space of rotation, and achieves competitive even when generalizing to unseen type of rotations in the z/SO3 setting. Methods without rotationally equivariant design (Wang et al., 2019; Zhang et al., 2019b; Thomas et al., 2019) depend primarily on augmenting training with rotations, allowing it to achieve competitive performance on easier rotations, but performing very poorly when generalizing to unseen $SO3$ rotations. Methods like RI-GCN (Kim et al., 2020) rely on designing invariant features at input to achieve rotational invariance, and so performs well when generalizing to unseen rotations. However, their performance is not best in z/z and SO3/SO3 settings, suggesting some loss of expressiveness from enforcing strict rotational invariance throughout the model. SFCNN uses rotationally equivariant convolutions based on icosahedral discretization, similar to our work. Unlike our approach, SFCNN is based on a single-sphere spatial representation, and uses a PointNet-like module Qi et al. (2017a) to learn point cloud projection. While competitive to our approach in z/z and SO3/SO3 settings, our approach generalizes much better to unseen rotations.

## 4.2 Resolving Electronic Structure of Materials

Accurate molecular dynamics simulation from quantum-mechanical principles is critical to many applications, such as the design of advanced materials or the study of materials' properties under extreme conditions. However, accurately scaling simulations to systems beyond hundreds of atoms is a problem of primary concern. The main bottleneck appears when solving the quantum mechanical questions, which provides properties describing the electronic structure, such as the electronic density of states (DOS) and the electron density. Recent ML efforts have tried to overcome this issue by effectively predicting the electronic structure (output) from the atomic structure (input) (Chandrasekaran et al., 2019; Fabrizio et al., 2019; del Rio et al., 2020; Kamal et al., 2020; Ellis et al., 2021). Here we aim to effectively and accurately predict the DOS, which describes the energy distribution of the electrons within an atomic snapshot. The DOS is a physical quantity that is invariant to rotations of the system. From the DOS, the band energy, an essential component of the total energy of the system, can be calculated. Due to the atomic nature of the problem, we propose learning atom-centered descriptors of local environments end-to-end, enabling data-driven and more

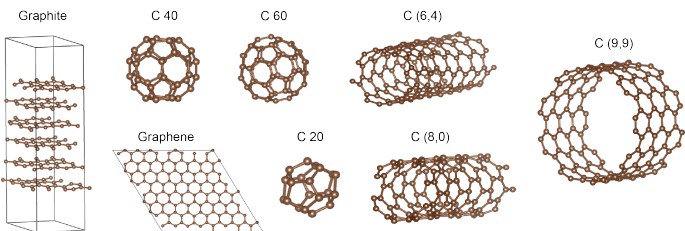

Figure 5: Illustrations of each type of carbon structure present in the dataset. They fall under three geometric classes: sheet-like (graphene, graphite), cylindrical (C(6,4), C(9,9), C(8,0)), and spherical (C20, C40, C60).

flexible representations compared to hand-crafted descriptors of prior work of del Rio et al. (2020); Ben Mahmoud et al. (2020). We further introduce and make publicly available a dataset consisting of geometrically diverse structures of graphene, and show that the use of CSGNN lowers overall error for calculating band energy, and increases the number of structures resolved to to chemical accuracy.

**Dataset.** The datasets consists of eight different types of graphene allotropes: graphene sheet, graphite, three different fullerenes, and three single-walled nanotubes–see Fig. 5). There are 200 atomic snapshots per structure, generated from snapshots of DFT molecular dynamics simulations run using VASP (Kresse & Furthmüller, 1996a;b). The number of atoms per snapshot range from 20 up to 152. After alignment with respect to the vaccum energy used as global reference, the resulting DOS curve is binned into 310 windows of 0.1 eV each, from -30 to 1 eV. We refer to del Rio et al. (2020) for more details on the data generation and preparation process. Using the same 80:20 split for each type of graphene structure results in 1280 total files for training and 320 files for validation A separate test set of size 160 is used with 20 snapshots per structure.

**Problem Formulation.** Each input is a snapshot of positions of carbon atoms, represented by coordinates $X \in \mathbb{R}^{N \times 3}$ and bounding lattice vectors $C \in \mathbb{R}^{3 \times 3}$. The atoms are located inside a unit cell, with periodic boundary conditions. The prediction target is the DOS, represented by a fixed dimension vector $y \in \mathbb{R}^{310}$. From the DOS an important downstream property of interest can be computed, the band energy:

$$E_{band} = \int_{-\infty}^{E_F} DOS(\epsilon)\epsilon d\epsilon \tag{4}$$

where $\epsilon$ is the energy and $E_F$ is the Fermi level. In practice, the integral is evaluated as a cumulative sum over a discrete and bounded domain. The integration has an upper bound of $E_F$, a physical limit representing the highest energy of the bound electrons. $E_F$ is calculated as the energy at which the cumulative integral of the DOS curve equals the total number of electrons in the system. Since this limit is a function of the DOS integral and electron number, we introduce an additional prediction target in terms of the cumulative DOS (FDOS). Including the FDOS during training enables better resolution of $E_F$ and lowers band energy error. The resulting objective function to minimize is:

$$L = \alpha * L_{DOS}(y, \hat{y}) + (1 - \alpha) * L_{FDOS}(y, \hat{y}) \tag{5}$$

where $\hat{y}$ is the predicted DOS and $\alpha$ controls the relative weighting between DOS and FDOS losses, which are both in terms of mean squared error.

**Model.** To model the total DOS, we use a decomposed approach to predict the contribution of each atom to the overall DOS. We further assume that each atom's contribution is a function of its local atomic atomic environment, an observation often exploited in electronic structure methods (Behler, 2016). The closer the neighboring atoms are to the target, the stronger the effect they have on the target's properties. To account for this effect, a fixed cutoff radius of 7 angstroms is used in experiments, eliminating the effect of neighbors that are further away. Our approach is then applied to learn a suitable descriptor of each local environment for mapping to atom-wise DOS contributions, which are then summed to obtain the overall DOS. This workflow is further illustrated in Fig. 6. Each atom serves as the center for concentric spherical representation, and the atomic neighboring environment is a point cloud. To convert a point cloud to concentric spherical feature map, each point is assigned a contribution to its nearest vertex. The contribution is determined as the inverse of the point's distance from center, based on the aforementioned neighbor effect.

All concentric spherical feature maps are combined channel-wise into a single group, resulting in single spatial sphere $R = 1$ with $C = 32$ channels. This eliminates effectively eliminates inter-sphere

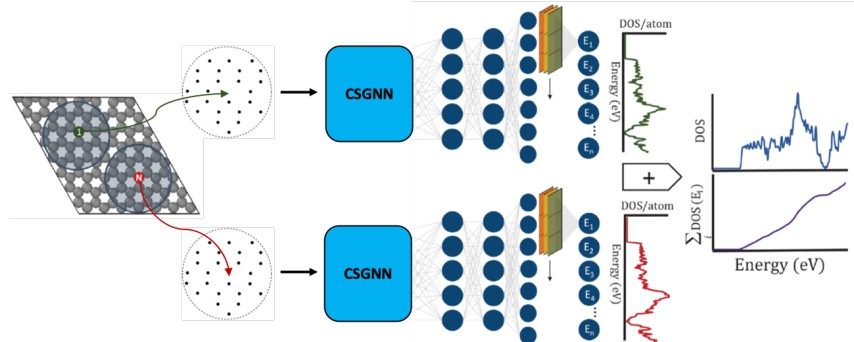

Figure 6: From left to right: each atom's local atomic environment is a point cloud, and the input for our proposed model. CSGNN (shared across inputs) learns a descriptor as output, which is mapped through additional dense layers and 1D convolution to predict atom-wise DOS contribution. All atoms' contributions are summed to obtain total DOS.

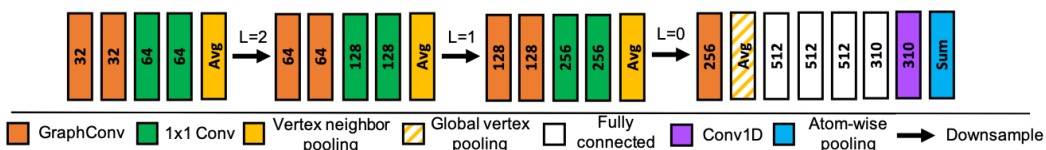

Figure 7: Architecture for DOS prediction. Number of output channels are shown for each layer, where applicable. $L = 3$ is initial icosahedral spherical resolution. 1x1 convolution is applied within channels, without spatial component. 1D convolution is applied to smooth the predicted DOS curve, represented by output energy bins.

convolutions, which instead become 1x1 convolutions in feature space. The main consideration here is computational and memory efficiency, as capturing co-radial information via input channels does not require adding additional spatial dimensions. We note that this version of our proposed architecture is already sufficient to obtain strong results in this problem. Fig. 7 presents the layers of the model architecture, and Sec. A.3 provides training and hyperparameter details.

**Results.** We present our results in Table 2. AGNI (Botu & Ramprasad, 2015; Botu et al., 2016) is a hand-crafted descriptor method for extracting rotationally invariant features of atomic environments applied to this problem and dataset in del Rio et al. (2020). SchNet (Schütt et al., 2017) is a neural message passing model for learning atom-centered features end-to-end, and is considered a strong baseline for many atomistic ML problems. Our rotationally equivariant approach achieves the lowest mean error in resolving band energy on the test set, reducing overall error by 24% relative to previous best. Our approach also demonstrates the ability of learned descriptors to improve over the performance of hand-crafted descriptors for this problem. We also included a single-sphere version of our model ($C = 1$) for ablation, which did not perform as well as the multi-sphere version ($C = 32$). Since the dataset is composed of different types of Graphene geometries, we further group test error by each type of structure. Our approach achieves the lowest mean error on six out of eight structures. Furthermore, our approach achieves chemical accuracy on more structures–chemical accuracy (0.043 eV/atom) is a widely adopted accuracy threshold in computational chemistry. We achieve chemical accuracy on seven structures, compared to six for SchNet and five for AGNI, out of eight.

### 4.3 ABLATION STUDY

In this section we further analyze some key components to the performance of our model, based on the model architecture for ModelNet40 classification task. We refer to Table 3 for all results.

**Number of concentric spheres.** We study the impact of concentric spheres on model performance. One option is to maintain a single sphere spatially ($R = 1$), and group all radial information from concentric spheres into input channels ($C$). Inter-sphere convolution is not applicable in this case, and only intra-sphere convolution is used. While increasing $C$ in this manner is effective, adding concentric spheres spatially is needed to achieve best performance, as seen in Table 3. The number of

| Method | Type | Parameters | Mean Error (eV/atom) |
|---|---|---|---|
| AGNI (Botu & Ramprasad, 2015) | Hand-crafted | 392K | 0.033 |
| SchNet (Schütt et al., 2017) | Learned | 976K | 0.035 |
| CSGNN ($R = 1, C = 1$) | Learned | 1.06M | 0.036 |
| CSGNN ($R = 1, C = 32$) | Learned | 1.06M | **0.025** |

(a) Overall error in resolving band energy for the test set. The mean absolute error per sample is normalized by number of atoms. $R$ is number of spatial concentric spheres, $C$ is number of concentric spheres as input channels.

| Model | Graphene | Graphite | C20 | C40 | C60 | C(6,4) | C(9,9) | C(8,0) |
|---|---|---|---|---|---|---|---|---|
| AGNI | 0.021 | 0.053 | 0.052 | **0.030** | **0.010** | 0.046 | 0.027 | 0.026 |
| SchNet | 0.042 | 0.045 | 0.065 | **0.030** | 0.022 | 0.022 | 0.033 | 0.019 |
| Ours (CSGNN) | **0.013** | **0.039** | **0.051** | 0.033 | 0.017 | **0.020** | **0.014** | **0.015** |

(b) Band energy mean absolute error of predictions for each type of structure. $C = 32$ version of our model is used here.

Table 2: Comparison of descriptors for resolving band energy from predicted density of states. Table 2a shows overall error over test dataset. Table 2b further categorizes energy prediction error by structure. CSGNN achieves lowest overall error, as well as lowest error in six out of eight structures.

| Setting | SO3/SO3 | Setting | SO3/SO3 | Setting | SO3/SO3 |
|---|---|---|---|---|---|
| *Spheres, channels only* | | *Spheres, spatial* | | *Radial convolution kernel size* | |
| $R = 1, C = 1$ | 84.1 | $R = 4, C = 8$ | 88.9 | $R = 16, K_{RC} = 1$ | 86.3 |
| $R = 1, C = 8$ | 87.6 | $R = 8, C = 8$ | 89.6 | $R = 16, K_{RC} = 3$ | **90.0** |
| $R = 1, C = 16$ | **87.9** | $R = 16, C = 8$ | 90.0 | $R = 16, K_{RC} = 5$ | 89.9 |
| $R = 1, C = 32$ | 87.7 | $R = 32, C = 8$ | **90.1** | | |

Table 3: Ablation results in terms of SO3/SO3 accuracy on Modelnet40 dataset. $K_{RC}$ is radial convolution kernel size. $C$ to refers to concentric spheres grouped via input channels, while $R$ refers to number of spheres represented spatially.

trainable parameters is same when increasing the spatial dimension, and so performance differences can be attributed to increased representational power via concentric spheres.

**Radial convolution.** We further provide justification for using the proposed radial convolution for learning features between spheres. The radial kernel size refers to the number of co-radial vertices considered in the convolution window. We set $K_{RC} = 1$, which operates over only the feature dimension and ignores the spatial dimension (analogous to 1x1 convolution for 2D images). This gave significantly worse performance than $K_{RC} = 3$, confirming that non-trivial radial convolution is necessary for performance. We did not observe any benefit to increasing kernel size beyond $K_{RC} = 3$.

## 5 CONCLUSION

To address the problem of learning rotationally robust representations of point cloud data, we propose a new volumetric feature learning approach. We propose a new volumetric structure based on nested spheres formed by the icosahedral discretization. We design intra-sphere and inter-sphere convolutions for learning over the concentric spheres, combined into a multi-resolution convolutional architecture. The proposed approach is both rotationally equivariant and efficient, scaling linearly with grid resolution. Our method achieves state-of-the-art performance in benchmarks for classifying arbitrarily rotated 3D objects. Our method is also effective in molecular environment description, and is applied towards more accurately resolving electronic structure of materials. An avenue for future work is extending our approach to variable icosahedral resolution across concentric spheres, in order to balance the resolution requirements of inner vs. outer spheres.

## REPRODUCIBILITY STATEMENT

To aid in the reproducibility of our work, we have taken the following measures:

- We have made our model and experiment implementations publicly available at https://github.com/anonymous10521/CSGNN. This also includes access to the datasets used in our experiments, from the repository.

- We have also provided pre-trained versions of our models used in reporting best results, in the repository.

- The repository includes instructions for setting up the environment and dependencies needed to run our implementation.

- To facilitate reproducing results from training, we have included relevant hyperparameters in Sec. A.2 and A.3 of the Appendix. We have also set default parameters of training scripts to match those in the paper, as much as possible.

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

# A  APPENDIX

## A.1  EQUIVARIANCE OF INTRA-SPHERE AND INTER-SPHERE CONVOLUTIONS

We define the feature map $h : S^2 \times \mathbb{R} \to \mathbb{R}$ as the function which maps positions on the spherical volume to scalar features (without loss of generality). A rotation $R$ on the feature map is then defined by the following operator:

$$[L_R h](\boldsymbol{x}) = h(R^{-1}\boldsymbol{x}) \tag{6}$$

A convolutional layer $\Phi$ is *equivariant* to the rotation group $\mathcal{R}$ if it commutes with all rotations:

$$\Phi[L_R h](\boldsymbol{x}) = [L_R(\Phi h)](\boldsymbol{x}), \quad \forall R \in \mathcal{R} \tag{7}$$

In other words, applying convolution over the rotated feature map is the same as applying rotation over the result of convolution. In this work we focus on equivariance to the icosahedral rotation group $\mathcal{I}$, a subgroup of $SO(3)$ containing 60 discrete rotational symmetries.

We start with intra-sphere convolution implemented by graph convolution within each sphere:

$$\Phi h(\boldsymbol{x}_{i,j}) = \sigma(\theta \sum_{\boldsymbol{x}_{i,k} \in N(\boldsymbol{x}_{i,j})} h(\boldsymbol{x}_{i,k})) \tag{8}$$

where $\boldsymbol{x}_{i,j}$ is the position corresponding to a vertex in the concentric spherical grid, indexed by radial and spherical dimension. Additionally, $\theta$ is trainable parameter, $N(\boldsymbol{x}_{i,j})$ denotes positions of vertices in the neighborhood of the vertex at $\boldsymbol{x}_{i,j}$, and $\sigma$ is nonlinearity function. Equivariance of the proposed graph convolution layer is shown as follows:

$$\Phi[L_R h](\boldsymbol{x}_{i,j}) = \Phi h(R^{-1}\boldsymbol{x}_{i,j}) \tag{9}$$

$$= \sigma(\theta \sum_{\tilde{\boldsymbol{x}}_{i,k} \in N(\tilde{\boldsymbol{x}}_{i,j})} h(\tilde{\boldsymbol{x}}_{i,k}), \quad \tilde{\boldsymbol{x}}_{i,k} = R^{-1}\boldsymbol{x}_{i,k} \tag{10}$$

$$= [L_R(\Phi h)](\boldsymbol{x}_{i,j}) \tag{11}$$

The second equality follows from rotation $R \in \mathcal{I}$ being an isometric transformation that maps the icosahedral sphere onto itself. This means that each vertex position $\boldsymbol{x}_{i,j}$ of the rotated feature map corresponds to an vertex unique position $R^{-1}\boldsymbol{x}_{i,j}$ in the original feature map, and that vertex neighborhoods are also preserved. The final equality follows from applying Eq. 6 and Eq. 8.

Next, we show that the intra-sphere convolution layer is also rotationally equivariant. Intra-sphere convolution is defined via 1D convolution over co-radial vertices:

$$\phi h(\boldsymbol{x}_{i,j}) = \sigma(\sum_{k=-\lfloor \frac{K}{2} \rfloor}^{\lfloor \frac{K}{2} \rfloor} h(\boldsymbol{x}_{i+k,j})\beta_{k+\lfloor \frac{K}{2} \rfloor}) \tag{12}$$

where $K$ is the size of the radial kernel and $\beta$ denotes trainable parameter. We show that the convolution $\phi$ commutes with rotation:

$$\phi[L_R h](\boldsymbol{x}_{i,j}) = \phi h(R^{-1}\boldsymbol{x}_{i,j}) \tag{13}$$

$$= \sigma(\sum_{k=-\lfloor \frac{K}{2} \rfloor}^{\lfloor \frac{K}{2} \rfloor} h(\tilde{\boldsymbol{x}}_{i+k,j})\beta_{k+\lfloor \frac{K}{2} \rfloor}), \quad \tilde{\boldsymbol{x}}_{i+k,j} = R^{-1}\boldsymbol{x}_{i+k,j} \tag{14}$$

$$= [L_R(\phi h)](\boldsymbol{x}_{i,j}) \tag{15}$$

The second equality follows trivially from the fact that co-radial vertices remain co-radial after shared rotation, thereby preserving neighborhood for convolution. Equivariance of the intra-sphere and inter-sphere convolutional layers ensures that compositions of these layers are also equivariant to rotation.

| Concentric spheres ($R$) | 1 | 8 | 16 | 24 | 32 |
|---|---|---|---|---|---|
| Training (seconds/epoch) | 51 | 83 | 104 | 150 | 204 |
| Inference (milliseconds/batch) | 70 | 75 | 97 | 119 | 155 |

Table 4: Training and inference time comparison for ModelNet40 using batch size 32, while varying the number of spatial spheres. Training time reported is seconds per epoch, while inference time is milliseconds per batch.

## A.2 MODELNET40 EXPERIMENT DETAILS

We specify CSGNN-z as the model trained on $z$-axis aligned rotations, and CSGNN-SO3 as the model trained on $SO3$ rotations. For concentric spheres, CSGNN-z uses 15 spheres spatially, and CSGNN-SO3 uses 20. Level 4 icosahedral discretization (2562 vertices) is used for initial spatial resolution of each sphere. Point clouds are mapped to vertex features using Gaussian RBF with of threshold $T = 0.01$. After vertex grouping, the number of concentric spheres as input channels is 8. Each model is trained using Adam optimizer for 60 epochs and batch size 32, with early termination if learning rate falls below 1e-5. CSGNN-z is trained with initial learning rate of 3.9e-4, and CSGNN-SO3 with initial learning rate of 2.2e-4. For regularization, CSGNN-z uses dropout of 0.19, and weight decay of 3.2e-7. CSGNN-SO3 uses dropout of 0.14, and weight decay of 2.7e-7. Beyond rotational augmentation in training, we apply random uniform translation in the range of $[-0.1, 0.1]$, random Gaussian noise for positional jitter with standard deviation of $0.01$, and random uniform re-scaling by a factor of $[0.8, 1.2]$ applied independently to each axis. For the z/SO3 test setting, it was beneficial to additionally apply voting over 5 sampled test rotations per instance.

## A.3 ELECTRONIC STRUCTURE OF MATERIALS EXPERIMENT DETAILS

We refer to the CSGNN model for density of states prediction as CSGNN-DOS. The initial spherical resolution is $L = 3$ icosahedron, or 642 vertices per sphere. CSGNN-DOS is trained using Adam optimizer for 1500 epochs, with initial learning rate of 5e-4 and early termination if learning rate falls below 1e-5. A batch size of 32 snapshots is used, and a weight decay of 1e-7 is applied for regularization. Using $\alpha = 0.1$ provided best performance for band energy calculation, for weighting DOS and FDOS losses. A single 1D convolution layer with kernel size of 3 is applied over the atom-wise output channels, corresponding to each atom's predicted DOS contribution, prior to summing to obtain the total DOS.

## A.4 RUNTIME ANALYSIS

We report how training and inference time scales with the addition of spatial concentric spheres to our model for the ModelNet40 task, in Table 4. The number of input channels is fixed at $C = 8$. Training time is seconds/epoch, averaged over the course of training. Inference time is milliseconds/batch, averaged over a run over the validation set. The training time increases from single sphere to 32 spheres by a factor of four, while the inference time increases by a factor of two. We also did not observe any benefit to model accuracy going beyond $R = 20$ for the ModelNet40 task in practice. Numbers were obtained from running on a single NVIDIA V100 GPU with 16 GB memory.

## A.5 INTRA-SPHERE POOLING

Pooling is widely used alongside convolutional filters in CNN architectures to learn invariance to transformations of the input. The icosahedron, due to its recursive refinement by discretization level, has a well-defined and natural hierarchy for pooling and coarsening. Each coarser representation also follows a uniform discretization of the sphere, which allows efficient information propagation in hierarchical fashion when combined with convolutions and pooling. To formalize the pooling operation, we introduce the overloaded notation $\mathbf{H}^{(l)} \in \mathbb{R}^{R \times |V^{(l)}| \times C}$ to denote the feature tensor corresponding to $V^{(l)}$, the vertex set corresponding to level $l$ discretization. Pooling is defined as $\mathbf{H}_{i,u}^{(l-1)} = f(\{\mathbf{H}_{i,v}^{(l)} : v \in N(u)\})$, where $N(u)$ is the neighborhood of vertex $u \in V^{(l)}$ and $f$ is a permutation invariant function (e.g. max operator). Pooling is followed by downsampling, where

| Structure | Graphene | Graphite | C20 | C40 | C60 | C(6,4) | C(9,9) | C(8,0) |
|-----------|----------|----------|-----|-----|-----|--------|--------|--------|
| Atoms | 128 | 108 | 20 | 40 | 60 | 152 | 144 | 128 |

Table 5: Each carbon structure, and the number of atoms for each snapshot of the structure.

only vertices of the smaller vertex set $V^{(l-1)}$ are retained. We only apply neighborhood pooling within vertices of the same sphere.

### A.6 POINT CLOUD TO VERTEX FEATURES

Instead of computing the summation in Eq. 1 with respect to all points, for each data point we update the features of vertices in a local neighborhood. The radial basis function (RBF) $\phi$ decays exponentially, and so points beyond a local neighborhood have little influence (depending on choice of bandwidth $\gamma$). Furthermore, restricting to a constant size local neighborhood reduces computation cost from $O(N|V|)$ to $O(N)$. To define the local neighborhood of data point $p$ in this work: any given point $p$ is contained within two bounding "triangles" of the discretization (ignoring boundary conditions and degenerate cases). These correspond to the vertices $S^{(i)} = \{u^{(i)}, v^{(i)}, w^{(i)}\}$ and $S^{(i+1)} = \{u^{(i+1)}, v^{(i+1)}, w^{(i+1)}\}$, where $i$ indexes radial level. However, using a single $\gamma$ value for the RBF results in scaling inconsistency: distances between vertices progressively shrink moving to inner spheres. Based on the assumption that RBF values should be invariant to scale, a different $\gamma$ and corresponding RBF is defined with respect to radial level. To define $\gamma_i$, we use the maximum pairwise distance $d_{max}^{(i)}$ between vertices in $\{S^{(i)}, S^{(i+1)}\}$. Specifically, we set $\gamma_i = \frac{-\log T}{d_{max}^{(i)\,2}}$, where $T$ is a lower bound target RBF value. For example, $T = 1$ would correspond to $\gamma_i = 0$, or a RBF value of 1 at any distance. $T \in (0, 1]$ is a tuning parameter that enables toggling the overall sensitivity of the RBF to distances. Based on the approximation that $d_{max}^{(i)}$ is similar for any data point, $d_{max}^{(i)}$ is precomputed once.

### A.7 ELECTRONIC STRUCTURE DATASET

We present the number of atoms per structure in Table 5.

