# OpenReview forum: "Concentric Spherical GNN for 3D Representation Learning"
_ICLR.cc/2022/Conference — ICLR 2022 Submitted_

### Official Review · Reviewer_vMzd · 2021-11-02

**Correctness:** 3
**Technical Novelty And Significance:** 2
**Empirical Novelty And Significance:** 1
**Recommendation:** 5
**Confidence:** 4

**Main Review:**

Strengths:
1. motivation: the motivation to use multiple resolution spheres with varying radius sounds valid and may help feature learning of 3D shapes, comparing to previous approaches using single unit sphere;
2. architecture: the multiple resolution approach is widely used in image deep learning architecture, this is the first time it is extended to spherical cnn and the proposed architecture could lead to many follow-up work in both 3D point cloud processing and spherical CNNs.

Weakness:
1. rotation-equivariant: authors claims the proposed method to be rotation equivariant and results on ModelNet40 also support this claim. This is the most important property of the proposed method. It is advisable to have a better explanation of both proposed method and related method mentioned in related work, i.e. why some methods are not rotation-equivariant.
2. ModelNet40: results of [1] (Cohen etal. 2019) is missing in Table 1.
3. DOS experiments:
(1). meaning of parameters R/C: in the experiments, R=1 and C=32 are used. Does this mean 32 spheres are combined into 1?
(2). loss functions: from A.2 and Eq.5, \alpha=0.1 is used, does this mean loss(DOS) is less important than loss(FDOS)? If previous works using \alpha=1.0, could the authors provides the results using the same setting? In other words, how does disabling loss(FDOS) hurt?
I'm also keen to know if it is fair to compare to other methods using additional loss term.
(3) Effectiveness of the methods: On DOS experiments, authors use simplified version of the architecture (R=1) and reported strong performance than baselines. Does this mean that DOS estimation is too easy for the proposed method?
(4) Spherical CNNs baseline: Considering the baselines use in DOS experiments, nearly all SOTA spherical CNNs are not compared.


Ref:
[1] Gauge equivariant convolutional networks and the icosahedral CNN.

**Summary Of The Paper:**

The paper presents a concentric sphere representation and icosahedron-based spherical CNNs for 3D point clouds. The contributions are two folds. First, the concentric sphere representation enables learning features volumetrically. Second, two types of convolutions: intra-sphere and inter-sphere are combined towards rotation equivariant and scalable computations.

**Summary Of The Review:**

In summary, the paper is easy to read and follow. Description of the method is clear and complete. Experiments include one for 3D point cloud classification and one for DOS estimation. For the latter one, I'm not familiar with the experiment difficulty and details and thus could not evaluate the methods from the results. Overall, I appreciate the efforts of the work to propose multi-resolution version of spherical CNNs. The downside is that evaluations are limited, and some baselines are not compared.

---

> ### Author Response · Authors · 2021-11-22
> **Response to Reviewer vMzd**
>
> We thank reviewer vMzd for the feedback on our work, and are glad that the paper is clear to follow.
> We present responses to the reviewer's raised concerns below, which we hope also addresses the reviewer's reservations about evaluations and baselines.
>
> ***More explanation of rotational equivariance in this work and in relation to other work
> We have added an additional section in the appendix with more theoretical discussion on rotational equivariance of our method.
> We have also re-worked the related work discussion to provide more background on rotational equivariance and why some methods either do not satisfy it or fail to satisfy effectively.
> We refer the reviewer to revised Sec. A1 and Sec. 2 of the paper for addressing of their stated concern.
>
> ***Missing reference Cohen et. al. (2019) in Table 1
>
> We are aware of Cohen et. al. (2019), which is already referenced in our paper.
> However, the authors did not compare ModelNet40 in their paper and unfortunately their code is publicly unavailable.
> As their method in the paper also focused on spherical images instead of general point cloud data, we decided to not include this work in the ModelNet40 comparison.
>
> ***Questions on DOS experiments
>
> (1) R and C correspond to spheres represented spatially and as input channels, respectively. R=1 and C=32 means 32 spheres are combined into 1, by collapsing co-radial vertices into single vertex and concatenating their features (scalar this case).
>
> (2) Prior work such as del Rio et. al. (2020) also use the FDOS loss term.
> To facilitate fair comparison, we included FDOS loss for all baselines, and try a range of $\alpha$ to select the best one.
> We found that disabling the FDOS loss ($\alpha=1$) drastically increases band energy error.
> This is because the cumulative DOS is used to determine the Fermi level, the upper limit of integration when calculating the band energy (Eq. 4).
> Without the FDOS, the error on the Fermi level and consequently the band energy calculation can be too high.
>
> (3) Effectiveness of simplified version of architecture:
> For the DOS prediction problem, we hypothesize that is sufficient to incorporate radial information of points via input channels, hence the already strong performance of the $R=1, C=32$ setting compared to baselines.
> Since inter-sphere convolutions are not used in this setting, it maybe that learning angular relationships of the data (via intra-sphere convolution) is more important.
>
> (4) We did not directly compare with other Spherical CNNs in the DOS experiment, due to a gap in their methodology for handling general point clouds, especially in the molecular context.
> We propose the concentric spherical model to effectively incorporate radial information, which is the key missing piece to being suitable for modeling molecular environments, more so than the particular choice of Spherical CNN.
> We demonstrate this point in comparing the performance difference between $R=1, C=32$ and $R=1, C=1$ setting, where the latter is representative of a Spherical CNN without concentric spherical representation.
> Concentric spheres permit surpassing the performance of existing methods based on hand-crafted descriptors.

---

### Official Review · Reviewer_KVV8 · 2021-11-07

**Correctness:** 3
**Technical Novelty And Significance:** 2
**Empirical Novelty And Significance:** 2
**Recommendation:** 5
**Confidence:** 5

**Main Review:**

This paper proposes a new volumetric representation that consists of multiple nested spheres, each discretized by the icosahedral grid. It proposes to learn the 3D volumetric representation over concentric spheres by combining intra-sphere and inter-sphere convolutions. The proposed convolutions are rotationally equivariant, and also scale (near) linearly with grid resolution. It shows some experiments in 3D object classification, and resolving electronic structure of atomistic systems.

The paper overall is technical sound with some interesting results. The application in atomistic systems is interesting and seems new at least to the reviewer for the 3D point clouds.

The results in 3D object classification is not very detailed or sufficient. There are a lot of datasets I wished the paper can test. Also it doesn't reference sufficient papers in the 3D point clouds deep learning domain which is related and very relevant. e.g the KPConv paper should be referred. The shellnet paper which also uses concentric shells should be mentioned. It seems to me the authors might not be very familiar with the literature in this area. I would like to see the paper has side by side comparisons with some of these leading papers in some of the standard benchmark datasets.

**Summary Of The Paper:**

This paper proposed a concentric spherical representation of 3D space, formed by nesting spatially-sampled spheres resulting from the highly regular icosahedral discretization. It utilizes separate intra-sphere and inter-sphere convolutions over the resulting concentric spherical grid, which are combined into a convolutional framework for learning volumetric and rotationally equivariant representations over point clouds.

**Summary Of The Review:**

Overall the paper is sound and interesting with good results. The application in atomic system is interesting and new. However the paper doesn't do a good comparison with leading papers in this field thus it is hard to judge the effectiveness of the proposed method.

---

> ### Author Response · Authors · 2021-11-12
> **Response to Reviewer KVV8**
>
> We thank Reviewer KVV8 for finding our work overall sound and interesting. We would like to follow up on the reviewer's feedback on experiments and comparisons with related work, including a follow-up question we had:
>
> *** Referencing sufficient papers in 3D point clouds deep learning domain
>
> We are aware of other work such as KPConv and ShellNet in the broader point cloud literature.
> While performing well on regularly aligned point cloud instances, they do not address the problem of effectively generalizing to rotations, which is the main focus of this work.
> We have focused our comparison and discussion to the literature most relevant to handling the challenge of rotated data.
> That said, we are open to including references such as KPConv and ShellNet to more completely highlight the broader space of point cloud learning work.
>
> *** Comparison on more datasets
>
> The reviewer states insufficiency of 3D object classification results, and wanting to see comparative evaluation on more  datasets.
> We emphasize that in addition to 3D classification, we also performed evaluation for a new problem in atomistic machine learning, highlighting the applicability of the proposed approach across different domains.
> Could the reviewer provide clarification on what additional dataset(s) or comparison(s) would be needed for more convincing evaluation?

---

> > ### Author Response · Authors · 2021-11-22
> > **Follow-up on response to Reviewer KVV8**
> >
> > In the absence of further clarification on additional dataset(s) or comparison(s) to be made, we have added comparisons with KPConv and ShellNet as originally mentioned by the reviewer.
> > Their results for ModelNet40 are added to Table 1 of the paper.
> > Their performance is significantly behind that of our method in the z/SO3 and SO3/SO3 settings.
> > We believe this further illustrates that in the absence of rotationally equivariant model design, point cloud methods which are considered state-of-the-art in non-rotated settings, tend not be competitive in the general rotation setting.

---

### Official Review · Reviewer_oVKM · 2021-11-07

**Correctness:** 3
**Technical Novelty And Significance:** 2
**Empirical Novelty And Significance:** 2
**Recommendation:** 5
**Confidence:** 3

**Details Of Ethics Concerns:**

An ethics statement should be included.

**Main Review:**

PROS
- The paper makes a very sensible contribution that extends a variant of spherical CNN to be multiscale. As such, better accuracy is expected because the finer grained details can be better captured.
- The presentation is quite clear and I did not spot significant problems.
- The chosen experiments are quite relevant and span two different communities.

CONS

- I feel like the paper presents what I would call a 'multi-resolution spherical CNN'. In other words, the only addition to the known graph based spherical CNN is the notion of multiple concentric spheres. However, the experiments seem to distinguish between Spherical CNN and the CSGNN with R=1, C=1. This confuses me a little bit. Would it be possible to discuss the differences or can we recover Spherical CNNs as a special case of the proposed approach?

- 'Representation' is a fundamental aspect in 3D analysis. I would suggest the paper to be more careful or conscious with the use of the word 'volumetric'. First, why are volumetric features desirable to begin with? Second, I feel like the concentric spheres can better capture the narrow band around the point set while still leaving parts of the space unexplained. This is not necessarily a negative, as it might amount to avoiding the storage of unnecessary information. In that sense, the representation might fall somewhere between a point cloud and a dense volume - say a sweet spot maybe. I might also be wrong, but these points could be better described in the paper.

- The related work does not really discriminate the proposed approach from the others so as to position this method among the state of the art. In particular, it could be good to discuss what properties/goals are not met by the prior approaches. Regarding the related works, it is also good to mention the initial attempts to develop invariant point cloud classification networks:
* Thomas, Nathaniel, et al. "Tensor field networks: Rotation-and translation-equivariant neural networks for 3d point clouds." arXiv preprint arXiv:1802.08219 (2018).
* Zhao, Yongheng, et al. "Quaternion equivariant capsule networks for 3d point clouds." European Conference on Computer Vision. Springer, Cham, 2020.

- The paper seems to lack some details of the rotational invariance capabilities of the convolutions. Especially invariant/equivariant design of the radial convolutions carry importance as being the main contributions of this paper.

- The paper seems to decouple two convolutions (radial and angular). Would it be possible to have a single convolution fusing information from both dimensions? (e.g. just like volumetric 3D convolutions) Could the paper explain why such a design choice is better?

- The paper argues for a spherical distribution to represent the point cloud. However this introduces some discretization artifacts. So it is of wonder to see whether the benefit brought by concentric spheres outdoes the discretization error.

- As shown in Fig. 2, it is possible to obtain a subset of the spheres at certain radii. As far as I understand this is somewhat a pruning mechanism. Is this only hypothetical or does it also happen with real point clouds? Could we see an example?

- The spheres seem to have more structure than a simple graph. I feel like resorting to GNNs can yield some information loss. Have we ever thought about this aspect?

- Can the approach estimate the pose of the object or is invariance property all we get? In case the pose can be estimated, can we see a study on this?

- In Fig. 6, I notice an architectural/design difference compared to the point cloud one. In fact, this seems to have spheres anchored on local patches (an operation also applicable to point sets). Do we necessarily need this change? Can we cover both problems with similar architectures?

- I feel that some kind of a theoretical analysis on the minimum number of concentric spheres required for rich feature extraction would be a nice to have.

Minor Remarks:
- such the underlying -> such that the underlying
- challenging to invariance -> challenging to introduce invariance (maybe?)
- 'over over' -> over
- Are all point sets normalized to the unit ball? Can we hear some more details here?
- What about the connections in Fig. 5? Are those used or is the discretization same as the point cloud experiment?

**Summary Of The Paper:**

This paper proposes a graph convolution based multiscale spherical deep neural network. In particular, radial convolutions over concentric spheres of different radii are used in conjunction with the typical spherical convolution. A simple 5-layer architecture is presented to fuse the rich features extracted by both convolutions. Experiments on point cloud classification under arbitrary SO(3) transformations as well as predicting electronic state density of graphene allotropes, demonstrate the validity of the devised method.

**Summary Of The Review:**

Simple idea, clear writing, but lacks thorough discussions and evaluations.

---

> ### Author Response · Authors · 2021-11-22
> **Response to Reviewer oVKM, Part 1**
>
> We thank Reviewer OVKM for the detailed feedback and questions. We address as many points as possible, below:
>
> ***Limited novelty as multi-resolution extension to Spherical CNN
>
> We argue that the paper has larger contribution scope than extending Spherical CNN with multiple concentric spheres. A key piece to the performance of the proposed model is intra-sphere convolution involving co-radial vertices, which enables learning features between spheres. This piece is a primary difference from existing Spherical CNNs and constitutes a novel treatment for concentric spherical representation learning. We empirically tested its significance in Table 3 through the $R=16, K_{RC}=1$ setting, which is akin to only using graph convolutions over concentric spheres. This setting replaces the intra-sphere convolution kernel with trivial 1x1 convolution, leading to significant performance drop (3.7\%). We further clarify that R=1, C=1 can be viewed as recovering a special case of the graph-based Spherical CNN using our approach.
>
> ***Benefits of learning volumetric features
>
> We argue that our approach is primarily a volumetric representation based on the volumetric sphere. We record both the angular and radial distribution of point clouds via the proposed concentric spherical grid, which can be viewed as a 3D spatial sampling of the spherical volume.
> This information is not sufficient captured via a single sphere, as in standard Spherical CNNs, due to the need for a projection to the 2D manifold.
> There are two main benefits of learning volumetric features over the proposed discretization.
> The first is that the concentric grid imparts structure, whereas the raw input is unstructured.
> This enables the design of spatially-aware and rotationally-equivariant filters over the structure that learns relationships between features native to 3D.
> The second benefit is that the regularity of the volumetric structure also provides a well-defined hierarchy for learning multi-resolution features.
> This allows the design of efficient pooling and downsampling operations, even though the input point clouds may be irregular and non-uniform.
> We note that these are advantages also observed analogously in 2D convolutional architectures, which have proven very successful.
>
> ***Limitations of prior work
>
> Our primary contribution is a rotationally equivariant approach that more effectively bridges the performance gap between state-of-the-art on non-rotated vs. rotated point cloud data, compared to prior work.
> We have re-structured the related work section to more clearly highlight the limitations of prior work.
> We have also added a bit more background explanation to further motivate rotational equivariance as our basis for rotationally invariant model design.
> We refer the reviewer to the updated related work section and hope it clarifies limitations of prior work.
>
> ***Lack of details on equivariance of convolutions
>
> We agree with the reviewer the rotational equivariance properties of the proposed convolutions can be explained in more detail.
> To address this concern, we have added a detailed theoretical discussion on the equivariance of both graph and radial convolution in Sec. A1 of the appendix, and refer the reviewer to it.
>
> ***Decoupled intra-sphere and inter-sphere convolutions vs. combined convolution
>
> We considered the possibility of a single kernel combining intra-sphere and inter-sphere dimensions.
> However, we regarded decoupling intra-sphere and inter-sphere convolutions as a more suitable and flexible design for concentric spherical grid.
> The spatial dimensions of intra-sphere and inter-sphere are not the same -- intra-sphere resolution is much higher (up to 2 orders of magnitude in experiments).
> Separating convolutions allows flexibility to allocate more layers and parameters to intra-sphere convolution than inter-sphere convolution, if that is the right thing to do.
>
> *** Alternative to GNN
>
> We used graph-based convolution as it performs well in practice and is also efficient, scaling linearly with grid size.
> However, we agree that more expressive operators can be considered for intra-sphere convolution for future work, due to the regular nature of the icosahedral discretization.

---

> > ### Author Response · Authors · 2021-11-22
> > **Response to Reviewer oVKM, Part 2**
> >
> > *** Multiple centers for DOS problem vs. single center for point cloud classification
> >
> > The primary difference in the architecture for the DOS prediction is the use of multiple centers for concentric spheres, centered on each atom of the structure.
> > The reasons for this difference are two-fold:
> > (1) The locality of the surrounding atomic environment of each atom is essential to predicting its contribution towards the total DOS.
> > As this locality is relative to each atom (in terms of distance to neighbors), using a single center (such as the centroid) would lead to a loss or distortion of this critical information.
> > (2) Physically, the total DOS of the system scales with the number of atoms in the system.
> > A model trained on locally atom-centered spheres can trivially scale to inference of larger atomic systems, whereas the same is not possible for a model trained on a single center.
> > The other direction, using multiple centers anchored on local regions for point cloud classification (instead of single centroid), could be an interesting extension for future work.
> >
> > ***Addressing additional questions/concerns
> >
> > *Subset of spheres example*: We clarify that the subset of spheres in Fig. 2 (shown as a sector) is for purposes of cleaner illustration only.
> > In implementation we convert point clouds to the full grid of concentric spheres.
> >
> > *Pose estimation*: We did not develop our method with the problem of pose estimation in mind. This would have to be investigated further, and we consider it outside the scope of the current work.
> >
> > *Point set normalization:* Point clouds are normalized to the unit ball with respect to the centroid in the classification experiment.
> > We did not normalize to unit ball in the DOS experiment, instead using max atom neighbor distances (across the dataset) to determine limit of spheres.
> >
> > *Connections in Fig. 5*: The connections in Fig. 5 are for illustrations purposes only, and are not used in the model.
> > The discretization strategy is same as in the classification experiment.

---

### Official Review · Reviewer_xhPe · 2021-11-07

**Correctness:** 3
**Technical Novelty And Significance:** 3
**Empirical Novelty And Significance:** Not applicable
**Recommendation:** 6
**Confidence:** 3

**Main Review:**

Strengths:
1. The author proposes a new volumetric representation as the basis for convolutional learning, which consists of multiple nested spheres discretized by the icosahedral grid.
2. The proposed CSGNN integrates both intra-sphere and inter-sphere convolutions to learn volumetric representation over concentric spheres.
3. The proposed CSGNN is practical in classifying arbitrarily rotated 3D objects and is also effective in molecular environment description.

Weaknesses:
1. The architecture is not novel enough, which is only based on hierarchical graph convolutions followed by radial convolutions, and the performance is restricted by the size of radius, as illustrated in Fig. 1 and the experimental results on Table 2.
2. It is time-consuming and redundant to adopt all the points to calculate the contribution to vertices, the reviewer is curious about how to select the effective points or how to avoid the redundant information when converting the point cloud to concentric spheres. Are there any adaptive strategies other than thresholding on some scores?
3. There are some adjustable hyper-parameters, i.e., the threshold of RBF, the number of spatial concentric spheres, the input channels, the kernel size, etc. It is not clear whether the method is robust.
4. How to calculate the DOS and FDOS loss items, the author is expected to give explanations.
5. Experimental results in some aspects are not state-of-the-arts, and some failure cases should also be discussed if there exists.


**Summary Of The Paper:**

This paper addresses a challenging problem in 3D representation learning of point clouds, which aims to generalize the representations well to arbitrary orientations. The proposed CSGNN uses a convolutional framework to learn over concentric spherical feature maps, which is incorporated with both intra-sphere and inter-sphere information. The idea is interesting and reasonably well presented. In general, the proposed CSGNN is practical on spherical representation learning. However, there still remains some issues that should be addressed.

**Summary Of The Review:**

This paper presents a new volumetric representation to learn 3D representations. However, some technical details should be further discussed and analyzed. Strengths and weaknesses are both discussed on the proposed CSGNN in terms of data process, method design, and experimental settings and results. Therefore, the reviewer gives rating of 6: marginally above the acceptance threshold.

---

> ### Author Response · Authors · 2021-11-22
> **Response to Reviewer xhPe**
>
> We thank Reviewer xhPe for overall finding the proposed ideas interesting and practical.
> We have add the following comments in order to address as many remaining concerns as possible:
>
> ***Architecture novelty, performance restricted by radius size
>
> While hierarchical graph convolutions and radial convolutions are indeed key pieces of the proposed architecture, we also regard their rotationally equivariant design as well as proposing a new spatial structured based on concentric icosahedral spheres as key contributions of our work. We hope these points can also be considered in the overall novelty of our architecture.
>
> With regards to the size of the radial dimension, we performed a study in Table 3, which shows the performance improvement flattens between R=16 to R=32. This seems to suggest that the size of the radial dimension is not a limiting factor to performance, and that even a relatively small number of concentric spheres gives good performance.
>
> ***Concern about redundancy/inefficiency in converting point cloud to concentric spheres from using all points
>
> Eq. 1 expresses individual vertex contribution with respect to all points, for simpler introduction.
> However, in practice we only consider a spatially limited neighborhood of points with respect to a given vertex.
> Computation is therefore localized in practice, and the entire conversion process scales linearly with the number of points, which is further explained in Sec. 6 of the Appendix.
>
> ***Adjustable hyperparameters and unclear robustness
>
> We refer to the study of key hyperparameters such as the number of spatial concentric spheres and radial kernel size in Table 3 to help address questions on this aspect.
> In both cases we identify that performance flattens but does not significantly decrease upon increasing the size of the hyperparameter past a certain point.
> This suggests a degree of robustness in the range of these hyperparameters.
>
> *** How to calculate DOS and FDOS loss terms?
>
> We understand this point may not have been fully elaborated in the text. The DOS and FDOS are both 260 dimensional vectors. For each sample or atomic structure, the loss of the DOS is computed as the mean squared error between predicted and reference DOS, and the same for FDOS.
> The weighted sum for both DOS and FDOS is computed by using the $\alpha$ term. The final loss is then computed as the mean over all atomic structures used during training.

---

### Author Response · Authors · 2021-11-22
**Summary of changes to paper**

We thank the reviewers for their time and giving feedback, suggestions, and questions based on our work.
Beyond responses to individual reviews, here we note some of the more substantial revisions we've made to the paper.
We have also uploaded a revised version of the paper, and marked significant changes in red for easier access.

Summary of changes to main text:
1. Added theoretical justification of the rotational equivariance of our model, in Section A.1 of the Appendix.

2. Re-written Section 2 (related work) to better introduce background on rotationally equivariant design and more clearly highlight the advantages of the proposed approach over prior work.

3. Added additional baseline comparisons with KPConv and ShellNet in Table 1 of ModelNet40 experiment.

---

### Decision · Program_Chairs · 2022-01-20

**Decision:**

Reject

**Comment:**

This paper addresses the problem of learning representation of 3D point clouds and introduces an interesting approach of concentric spherical GNN with the property of rotationally equivariant. It shows some promising results on point cloud classification under SO(3) transformations and on predicting electronic state density of graphene allotropes. The reviews suggest that, while it does not suffer from any major flaws, the paper has a fairly large number of minor issues that add up to make it subpar for publication. The proposed approach have several hyperparameters, but the authors do not seem to be up front about how the parameters are selected except for stating that they use "standard tuning techniques" --- this is not a satisfactory answer and appears to be dodging the question. Many technical details and specific choices could use more thorough explanation and analysis. The distinction of the proposed approach in relation to the large body of existing literature could be more clearly spelled out. Collectively, these issues made the contribution of this paper less clear.